# Effects of Different Endometrial Preparation Regimens during IVF on Incidence of Ischemic Placental Disease for FET Cycles

**DOI:** 10.3390/jcm11216506

**Published:** 2022-11-02

**Authors:** Yingjie Wang, Qiuju Chen, Yun Wang

**Affiliations:** Department of Assisted Reproduction, Shanghai Ninth People’s Hospital, Shanghai Jiao Tong University School of Medicine, Shanghai 200025, China

**Keywords:** endometrial preparation regimens, ischemic placental disease, in vitro fertilization, pre-eclampsia

## Abstract

We conducted this retrospective cohort study aiming to compare the different pregnancy outcomes of endometrial preparation regimens on ischemic placental disease in a frozen embryo transfer cycle. The study included a total of 9351 women who had undergone therapy at our single tertiary hospital from January 2015 to July 2020. The women were divided into three groups depending on their endometrial regimens: natural cycle, stimulation cycle, hormone replacement therapy cycle. The data were analyzed after propensity score matching, then we used multiple linear regression to study the relationship between ischemic placental disease and endometrial regimens, adjusted by confounding factors including age, body mass index, and score of propensity score matching. We performed univariate logistic regression, as well as multivariate logistic regression for ischemic placental disease, small for gestational age infant, placental abruption. and pre-eclampsia, respectively, listing the odds ratio and *p*-values in the table. As a result, risk of ischemic placental disease and small for gestational age infant were detected as higher in stimulation cycles compared to natural cycles before or after adjustment. Hormone replacement therapy cycles conferred a higher risk of pre-eclampsia and preterm delivery compared to natural cycles. No difference was found between stimulation cycles and hormone replacement therapy cycles, regardless of whether they are adjusted or not. In summary, more pharmacological intervention in endometrial preparation was associated with a higher risk of ischemic placental disease related symptoms than natural cycles for endometrial preparation in women undergoing frozen embryo transfer. Our findings supported that minimizing pharmacological interventions during endometrial preparation when conditions permit has positive implications for improving pregnancy outcomes.

## 1. Introduction

Ischemic placental disease (IPD) is a series of symptoms that involves pre-eclampsia, intrauterine growth restriction or small for gestational age infant (SGA), and placental abruption [1,2]. There are some studies on the causes of pre-eclampsia, intrauterine growth restriction and placental abruption, but effective experience is still lacking. This may be due to the fact that the three diseases are not clinically treated together; thus, the consistency of their etiology is ignored. IPD is a collective concept of pre-eclampsia, small for gestational age infant, and placental abruption, the current studies support a common origin in their etiology, but the relevant clinical pathogenic factors have not been widely reported [3].

From an etiological perspective, the common mechanism for these complications may be that placental vascular implantation disorders occurs during the placental implantation stage, resulting in placental ischemia and placental insufficiency. Excessive superficial implantation of spiral arteries in the trophoblast is the main reason for the subsequent development of pre-eclampsia and intrauterine growth restriction [4,5]. It has been shown that the development of pre-eclampsia and intrauterine growth restriction depends on the depth of trophoblast invasion. These theories suggest that interference during the placental implantation stage will more likely lead to IPD.

In the past few decades, in vitro fertilization (IVF) has been applied to the treatment of infertility, which has been found to increase the risk of pre-eclampsia, SGA, and placental abruption alone [6,7,8,9,10]. Previous studies have also shown that the incidence of elevated progesterone, gestational hypertension, fetal growth restriction, perinatal bleeding, and congenital malformations is higher in IVF in singleton pregnancies [5,11]. This series of studies suggests that IVF is more likely to be accompanied by adverse obstetric outcomes than natural pregnancy. Some of the obstetric complications associated with IPD are also included. Therefore, whether IVF has a direct or indirect relationship with IPD has become our point of interest. One of the risk factors for IPD may be inappropriate medical interventions during IVF, which means that the likely source of risk needs to be identified and avoided in clinical practice [11].

Clinical studies on IVF and IPD are still in the exploratory phase [12]. Previous work by Johnson KM et al. showed that the risk of placental insufficiency occurring in frozen embryo transfer cycles is lower when compared to fresh embryo transfer cycles [13]. This association was largely due to the lower risk of placental insufficiency during the frozen embryo transfer cycle. Another study showed an increased risk of IPD after donating oocyte fertilization in vitro [14,15]. However, the effect of the protocol during endometrial preparation on IPD has not been reported. Given the large number of studies demonstrating the effect of the endometrial regimen on perinatal complications, we hypothesized that the endometrial preparation regimen was more strongly associated with IPD due to the application of multiple hormones in the in vitro fertilization cycle. Exploring the endometrial-dependent relationship between IVF and IPD will facilitate subsequent clinical practice to assess the relationship between IVF and placental insufficiency.

## 2. Purpose of the Work

The purpose of this study was to compare the effect of endometrial preparation protocols during the cycle of FET on the incidence of ischemic placental disease. Based on the experience of the previous study, only patients with FET cycles and single-fetal pregnancy were selected as a cohort to exclude the instability of fresh cycles and multiple pregnancies. According to previous studies, IPD is closely related to obstetric outcomes, so our observational indicators mainly focus on the gestational period [16]. In this single-center study, we performed a retrospective cohort study incorporating three representative treatments of the endometrial preparation phase, comparing pre-eclampsia, placental abruption, birth weight, gestational age (GA), and associated outcomes in women undergoing natural cycle FET, stimulation cycle FET, and HRT (hormone replacement therapy) cycle FET.

## 3. Method

### 3.1. Study Design and Population

The study was a retrospective cohort study conducted in the assisted reproduction department of Shanghai Ninth People’s Hospital affiliated with Shanghai Jiao Tong University School of Medicine. We selected women who received frozen thawed embryo transfer (FET) IVF/ICSI cycles from January 2015 to July 2020. Women meeting the following inclusion criteria were included: underwent FET cycles; successfully delivered a live fetus; singleton pregnancy. Exclusion criteria were disappearance of twin syndrome and congenital uterine malformation, as determined by ultrasound or hysteroscopy; at least one missing data [17]. Furthermore, only the first pregnancy was retained if the woman had more than one delivery during the study period. Obstetric and neonatal data were obtained from questionnaires sent to parents in the middle and late pregnancy and after the due date of delivery. Issues related to demographic data were completed by parents and obstetric/neonatal data were provided by the gynecologist and/or pediatrician. The study was approved by the institutional review board of the hospital.

In this study, the propensity score matching (PSM) statistics method was applied to correct for the confounding bias in the sample baseline in a 1:1 ratio. We used the endometrium preparation protocol as the dependent variable with the baseline difference term in the unmatched case as the predictor. Data from different subgroups were matched in pairs, thus clarifying the true outcome differences between different endometrium preparation protocols.

### 3.2. Endometrial Preparation Procedures

Endometrial preparation regimen includes three types: natural cycle, stimulation cycle, or HRT alternative cycle regimen. As the current evidence shows that no regimen is absolutely dominant, each regimen was decided by the doctor in charge after considering the information about patient age, BMI, past medical history, and patient preferences.

During the natural cycle, we monitored the follicle growth by serum hormone and transvaginal ultrasound starting from day 10 of the cycle. It was monitored regularly every 2 days. The HCG was used to trigger the ovulation. The time of embryo transfer was determined based on the embryo freezing time and LH level. If the LH level was <20 IU/L when the LH was spontaneously surging, 5000 IU HCG was injected in the 9:00 p.m. when the dominant follicles were scanned, and the thawed embryo transfer was arranged 4 days later. Similarly, thawing and transfer of blastocysts was scheduled on day 6 or day 7, based on serum hormone levels and sonographic findings, according to the same criteria. Luteal support was performed from day 3 after HCG injection. Duphaston (40 mg/d; Abbott Bio, Chicago, IL, USA) wasused for luteal support [18].

Stimulation cycles are usually applied to patients with irregular menstrual cycles. The common method is to stimulate the single follicle growth by oral letrozole 2.5–5 mg daily from days 3 to 7, and to monitor the follicle growth starting from the 10th day of the cycle. If the dominant follicle diameter was <14 mm, patients received an additional 150 IU of HMG every 2 days to stimulate both follicle and endometrial growth. If the dominant follicle was >14 mm in diameter, the time given with 10,000 IU HCG and FET was performed using the same criteria described above [19].

HRT cycle, also known as hormone replacement therapy, is used in patients with a history of endometrial disease during the natural cycle or stimulation cycle. Oral E2 began on the second or third day of a natural or progesterone-induced menstrual cycle. Once the endometrial thickness was >8 mm, or with E2 administration for 42 days, 4 yellow hemostatic tablets (total E2 8 mg and anti-progesterone 40 mg; solid pharmaceutical) per day were taken [19]. Embryos thawed and transplanted were scheduled on day 3 or 5 after cutting stage embryos or blastocyst injection, respectively. After 14 days, ultrasound was performed to measure endometrial thickness to ensure that no dominant follicles appeared. When the endometrial thickness was >8 mm, vaginal progesterone suppositories (400 mg/day; Utrogestan; Besins Healthcare, Brussels, Belgium) and yellow oral Fematon tablets (consisting of 2 mg 17β-oestradiol and 10 mg dydrogesterone per tablet, 6 mg/day) were applied. Once pregnancy occurred, luteal support continues gestation for 8–10 weeks [20].

### 3.3. Embryo Quality Assessment and Vitrification

If cleavage-stage embryos had been counted for at least six blastomeres, then they would be defined as high-quality embryos. Minor criteria include embryos fragmented by 20 percent, which had no significant morphological abnormalities. High-quality cleavage embryos were selected for vitrification, whereas less than high-quality embryos were longer cultured to blastocysts. All embryos were vitrification cryopreserved during the study period, a process that was performed with dimethylsulfoxide–glycol–sucrose as a cryoprotectant. Embryos were transferred to a dilution solution and thawed before embryo implantation. Detailed vitrification and thawing protocols have been reported in previous studies at our center. No more than two embryos were transplanted in all FET cycles [20].

### 3.4. Observational Indicators and Outcome Measures

The main observation of the present study is the IPD. Here, the IPD is defined as the following conditions: pre-eclampsia, placental abruption, and SGA. Pre-eclampsia was defined as the presence of high blood pressure (>140/90 mmHg) during gestation, and pre-eclampsia (headache, visual changes, severe abdominal pain), eclampsia, or abnormal laboratory indicators (proteinuria, alanine, transaminase >80 IU/L, or platelet >100,000) before delivery. Placental abruption is defined as evidence of abruption or a blood clot during delivery; evidence of placental pathological abruption; or a very strong clinical suspicion for hospitalization and intervention [13]. SGA infants were defined as birth weight below the 10th percentile adjusted for sex and gestational age [21]. Secondary outcome measures included singleton birth weight, gestational age (calculated from the ET day), LBW (<2500 g), LGA. The study outcomes also included preterm delivery (<37 weeks), very preterm delivery (<32 weeks), and very low birthweight (VLBW, <1500 g). Birthweight percentile calculations are all based on the Chinese national birth weight reference data. Diagnoses of pre-eclampsia and placental abruption were obtained by follow-up of patient records by reviewers who were not involved in the study. The follow-up system of our center has been described previously.

### 3.5. Statistical Analysis

All data were obtained using SPSS software version 25.0 (SPSS Inc., Chicago, IL, USA) and R software version 4.1.3. Baseline characteristics and perinatal outcomes within the paired groups were tested using the following rules: hypothesis tests were performed using the chi-square test, Mann–Whitney U test, or ANOVA, depending on the type of data, distribution, and sample size. A *p*-value <0.05 indicates statistical significance. Crude odds ratios and adjusted odds ratios are shown along with the corresponding *p*-value. We used the univariate Logistic regression to calculate the crude odds ratios. For women with singleton pregnancies, some studies have suggested that factors such as age and BMI have an effect on IPD-related diseases [22]. Thus, the adjusted odds ratios were based on maternal characteristics (age, body mass index [BMI]), and the PSM scores.

## 4. Results

### 4.1. Baseline Characteristics of Patients Involved

A total of 9351 cycles were analyzed in our center, when conducted between January 2015 and July 2020. Among them, 1678 natural cycles FET, 4017 stimulated cycles FET, and 3134 HRT cycles singletons were used. Following PSM, 1630 cycles of natural cycling FET with stimulated cycling FET, 1540 cycles of stimulation cycling FET with artificial cycling FET and 2823 cycles of natural cycling FET with artificial cycling FET were included in the analysis. There were no significant differences between the groups at baseline after matching. As expected, these features were comparable between groups; see Table 1.

### 4.2. Subgroups Analysis

Neonatal outcomes stratified by the type of endometrial preparation protocol are shown in Table 2, Table 3 and Table 4. Overall, neo-natal IPD-related criteria after stimulation cycle FET and HRT cycles were worse than those after natural cycle FET.

In the matched natural cycle FET and stimulated cycle FET groups, an increase in the odds of IPD and SGA was detected among the stimulated cycle FET group before or after adjustments for covariates (*p* < 0.05). No other indicators showed differences in the two data groups, either before and after regression correction. However, it is worth noting that most of the adverse outcome measures of the stimulated cycle were higher than those in the natural cycle except LGA, although there were no statistical differences. (Table 2).

In the matched natural cycle FET and HRT cycle FET groups, higher odds of pre-eclampsia and preterm delivery were detected both before (*p* < 0.05) and after (*p* < 0.05) regression correction, indicating significant differences (Table 3).

In the stimulated cycle FET and HRT cycle FET groups, no outcome measures showed statistical differences, although all the indicators were higher in the HRT cycle group than in the stimulated cycle group (Table 4).

## 5. Discussion

We conducted this large single-centered retrospective cohort study to provide evidence for clinical guidance. We found a lower risk of IPD from placental insufficiency among natural cycles compared with HRT and stimulation cycles in this cohort study of deliveries at a single center over a 5-year period. Lower risk of SGA, pre-eclampsia, and placental abruption in the natural cycle group likely drives this association. We found that the three factors were more common in stimulation cycles and HRT cycles, though these differences were not totally statistically significant.

The risk of developing pre-eclampsia in the HRT cycles was higher than in the natural cycles, and the extent of this effect was significantly higher than that observed in other studies when assessing the relationship between pre-eclampsia and the frozen embryo transfer cycle [23]. This may be highlighted because the incidence of pre-eclampsia is not high, and our large database includes more samples of this disease. In addition, preterm delivery was significantly higher than the natural cycle in both the stimulation cycle and the HRT cycle, which is consistent with previous studies. We pay particular attention to this indicator here; although the etiology of preterm birth is very heterogeneous, the link between IPD and preterm birth is particularly important, and their pathogenesis and clinical epidemiology are also highly correlated [11,24,25].

SGA is defined as a birth weight lower than the 90th percentile of the birth weight of a fetus born in the same gestation week and of the same sex. In this study, the incidence of SGA in natural cycle FET was lower than in HRT cycle FET or stimulated cycle FET, although there was no significant difference between natural and HRT cycles. Here, we focus on differences in hormone use between different groups. It is well known that uterine placental infusion increases the likelihood of fetal growth restriction, which increases the incidence of SGA. The early pregnancy period is a critical period for the construction of blood circulation between the mother and fetus, and the fluctuations of the internal and external environment can easily interfere with the production of blood vessels and blood perfusion. Therefore, we hypothesis that the difference in the incidence of SGA in people with different endometrial preparation protocols is caused by differences in the intensity of hormonal stimulation in the early gestation, while this conjecture needs further verification.

We found less probability of IPD in a natural cycle pregnancy compared to HRT and stimulated cycles. Most previous studies have shown that stimulation or HRT cycles are usually accompanied by higher complications or adverse effects, in which changes in hormone levels due to drug use may play a role. This is in line with our expectations. Previous studies have found an increased risk of stimulated cycles or HRT cycles for low birthweight or SGA as compared to natural cycles [20]. All cycles involved received similar control for ovarian hyperstimulation, so we hypothesized that the association of drug-provided cycles with IPD or SGA was due in part to fluctuations in hormones, except for essential, previously diagnosed vascular disease. Previous studies suggest that inappropriate use of hormones may affect angiogenesis, and thus, placental implantation, which most likely affects the incidence of IPD, although they did not specifically assess differences between endometrial regimens in IVF treatment [26,27].

The reason why endometrial regimens result in higher rates of IPD and related outcomes only in medicine used groups is likely to be complex. Previous studies have shown that hormones have an important impact on the development of the endometrium and placenta. Appropriate concentration of estrogen levels is conducive to angiogenesis, thus promoting placental development and fetal growth. However, high concentrations of estrogen can instead inhibit the migration and proliferation of endothelial cells, leading to abnormal development of the placental vascular network [28,29]. Therefore, more medical intervention, such as the application of hormone or steroid medicine, might change the biochemistry condition. Pharmacological modulation of therapeutic purposes is particularly challenging due to the lack of attention to the immune status of currently used drugs. Identifying the determinants of the biological response of the vascular immune interface to estrogen drugs and developing targeted pharmacological interventions may yield new and improved treatment options.

We hypothesize that HRT cycle FET increases the risk of IPD, mainly by fluctuating the hormone in a significant way. Mean birthweight and term gestation indicate no significance. This can be summarized as an HRT cycle having limited influences on maternal body, while it shows a lack of obvious effects on the newborns. These effects may receive less consideration compared with pathophysiological influences fetal growth, but more attention to the impact of hormone fluctuation on complications of the patients themselves.

Our results suggest that the natural cycle may be a relatively safer approach than stimulation or hormone replacement therapy, although it is unclear whether this is due to basic disease, cryopreservation techniques, or other features of the IVF cycle. Further studies on the relationship between the various parameters of the IPD and the IVF cycle, and on the interactions with essential diseases are needed to elucidate this relationship. From the data of this study, although there was no statistical difference between most schemes except IPD, pre-eclampsia, SGA, and preterm delivery, we can still see that the incidence of adverse outcomes of natural cycle FET is lower than that of stimulation cycle or hormone replacement cycle, so the natural cycle is still the better choice. Both the previous conclusions and the statistical results of this study support this viewpoint [30,31]. Further exploration of the differences in IPD endometrial protocols in IVF is essential both for understanding the underlying biology and for identifying targets for future research. IPD shares the same pathogenesis as pre-eclampsia, placental rupture, and SGA [15]. It is widely accepted that in some pregnancies affected by pre-eclampsia, placental development is disrupted, leading to cellular, molecular, immune, and vascular changes, and that the role of insufficient decidualization has also received increasing attention, especially in vascular pathology [32]. Intrauterine placental abnormalities and superficial trophoblast invasion can mediate incomplete spiral artery remodeling. This may lead to placental hypoxia, aberrant angiogenic status, endothelial dysfunction, further reduction in placental formation, trophoblast stress, and can ultimately lead to maternal pre-eclampsia and placental abruption. For molecular targets, less explored areas include regulatory RGS proteins that mitigate the negative effects of excessive GPCR induced by hormones such as angiotensin II, endothelin-1, and vasopressin, leading to mitochondrial dysfunction, cell death, circulating DNA, and subsequent cellular stress of TLR9 activation. Any change in the internal environment may cause a range of pathophysiological dysfunction. For pregnant women, the hormone level is undoubtedly the most important unstable factor [32,33].

A limitation of this study is that it is a retrospective study based on data from a single reproductive center. Limited to retrospective studies, self-reported data is inherent to cohort studies. Therefore, we used a propensity score-matching approach to adjust several indicators of different FET cycle regimens and improve differences in baseline patient characteristics which lead to endogenous differences. In addition, the study counted the data over a long period, and in some recent clinical cases, there were multiple changes in the specific medication regimen, especially the stimulation cycle, but the overall medication principles and observation indicators are unchanged, so the difference in this part is controllable. Although we cannot explain all changes in practice that may affect our findings, especially maternal lifestyle or medication use during pregnancy, and underlying conditions that were not able to observe prior to treatment, such as metabolic syndrome, we counted relevant factors and follow-up data whenever possible, as shown in Table 1.

The strengths of this study include the large sample size and the use of IVF and well-established follow-up records. This allowed us to examine the independent risk factors for IPD, identify potential confounders, and perform a regression correction to adjust for bias. The composite results of three individual pregnancy complications with a broad range of shared risk factors and the underlying common pathophysiology could facilitate the understanding of the underlying mechanisms, giving our study more attention. We hope to compare donor-based IVF versus autologous versus non-IVF pregnancies in the following study, which could preclude the effect of controlling for ovarian hyperstimulation during egg retrieval. This helps further our understanding of these mechanisms.

## 6. Conclusions

In conclusion, regardless of the statistically significant difference in general pregnancy outcomes, the incidence of IPD, placental abruption, pre-eclampsia, and SGA was significantly lower in natural cycle FET. When we adjusted for confounders, the differences in outcome between IPD and SGA was maintained compared with the stimulated cycle. Differences in preterm delivery and pre-eclampsia between natural cycle and HRT cycle also exist after adjustment. These results suggest that different endometrial preparation protocols have varying degrees of potential negative impact on pregnancy, although we cannot yet figure it out, and further research on this issue is warranted.

## Figures and Tables

**Table 1 jcm-11-06506-t001:** Baseline characteristics of women undergoing FET via the Natural cycle and Stimulation cycle/Natural cycle and HRT cycle/Stimulation and HRT cycle.

Characteristic	Natural Cycle	Stimulation Cycle	*p*-Value	Natural Cycle	HRT Cycle	*p*-Value	Stimulation Cycle	HRT Cycle	*p*-Value
	1630	1630		1540	1540		2823	2823	
AGE	36.569 (4.141)	36.369 (4.350)	0.154	36.552 (4.702)	36.539 (4.199)	0.796	35.670 (4.366)	35.633 (4.605)	0.358
Duration of infertility			0.479	3.17 (2.76)	2.95 (2.77)	0.989	3.07 (2.89)	2.95 (2.77)	0.996
0	282 (17.301%)	312 (19.141%)		289 (18.766%)	276 (17.922%)		553 (19.589%)	564 (19.979%)	
1	197 (12.086%)	208 (12.761%)		202 (13.117%)	189 (12.273%)		345 (12.221%)	350 (12.398%)	
2	301 (18.466%)	305 (18.712%)		289 (18.766%)	285 (18.506%)		511 (18.101%)	524 (18.562%)	
3	245 (15.031%)	261 (16.012%)		229 (14.870%)	230 (14.935%)		474 (16.791%)	467 (16.543%)	
3+	605 (37.117%)	542 (33.251%)							
BMI	21.450 (2.829)	21.458 (2.780)	0.471	21.490 (2.812)	21.481 (2.840)	0.647	22.178 (3.186)	22.193 (3.321)	0.630
Excellent embryo rate	0.520 (0.251)	0.514 (0.247)	0.546	0.528 (0.264)	0.526 (0.252)	0.794	0.505 (0.245)	0.509 (0.255)	0.852
Cyclerank			0.410			0.942			0.696
1	928 (56.933%)	925 (56.748%)		812 (52.727%)	823 (53.442%)		1264 (44.775%)	1314 (46.546%)	
2	428 (26.258%)	395 (24.233%)		441 (28.636%)	431 (27.987%)		915 (32.412%)	847 (30.004%)	
3	154 (9.448%)	176 (10.798%)		154 (10.000%)	157 (10.195%)		387 (13.709%)	394 (13.957%)	
3+	120 (7.362%)	133 (8.160%)		(%)	(%)		(%)	(%)	
Gravidity			0.079			0.215			0.689
0	916 (56.196%)	863 (52.945%)		818 (53.117%)	859 (55.779%)		1604 (56.819%)	1619 (57.350%)	
1	373 (22.883%)	409 (25.092%)		371 (24.091%)	354 (22.987%)		656 (23.238%)	633 (22.423%)	
2	213 (13.067%)	210 (12.883%)		190 (12.338%)	203 (13.182%)		328 (11.619%)	323 (11.442%)	
3	83 (5.092%)	83 (5.092%)		105 (6.818%)	80 (5.195%)		149 (5.278%)	161 (5.703%)	
3+	43 (2.638%)	65 (2.988%)		(%)	(%)		(%)	(%)	
Parity			0.759			0.602			0.497
0	1466 (89.939%)	1470 (90.184%)		1380 (89.610%)	1387 (90.065%)		2578 (91.321%)	2576 (91.250%)	
1	150 (9.202%)	149 (9.141%)		151 (9.805%)	140 (9.091%)		225 (7.970%)	232 (8.218%)	
2	13 (0.798%)	11 (0.675%)		9 (0.584%)	12 (0.779%)		18 (0.638%)	15 (0.531%)	
3	1 (0.061%)	0 (0.000%)		0 (0.000%)	1 (0.065%)		2 (0.071%)	0 (0.000%)	
Paternal factor	377 (23.129%)	380 (23.313%)	0.901	337 (21.883%)	340 (22.078%)	0.896	625 (22.140%)	604 (21.396%)	0.498
Ovulation Dysfunction	90 (5.521%)	67 (4.110%)	0.060	92 (5.974%)	90 (5.844%)	0.879	505 (17.889%)	507 (17.960%)	0.945
Endometriosis	142 (8.712%)	144 (8.834%)	0.901	150 (9.740%)	144 (9.351%)	0.713	208 (7.368%)	221 (7.829%)	0.514
Tubal disease	1157 (70.982%)	1151 (70.613%)	0.817	1087 (70.584%)	1076 (69.870%)	0.665	1763 (62.451%)	1762 (62.416%)	0.978
Unexplained infertility	27 (1.656%)	27 (1.656%)	1.000	21 (1.364%)	26 (1.688%)	0.462	49 (1.736%)	45 (1.594%)	0.677
No. of embryo transferred			0.902			0.709			0.616
1	396 (24.294%)	393 (24.110%)		394 (25.584%)	385 (25.000%)		794 (28.126%)	811 (28.728%)	
2	1234 (75.706%)	1237 (75.890%)		1146 (74.416%)	1155 (75.000%)		2029 (71.874%)	2012 (71.272%)	
Developmental stage of embryo			0.350			0.661			0.920
Day 5 or 6	265 (16.258%)	285 (17.485%)		248 (16.104%)	257 (16.688%)		560 (19.837%)	557 (19.731%)	
Day 3	1365 (83.742%)	1345 (82.515%)		1292 (83.896%)	1283 (83.312%)		2263 (80.163%)	2266 (80.269%)	
Insemination method			0.897			0.963			0.862
IVF	1014 (62.209%)	1002 (61.472%)		946 (61.429%)	952 (61.818%)		1612 (57.102%)	1623 (57.492%)	
ICSI	488 (29.939%)	500 (30.675%)		468 (30.390%)	461 (29.935%)		818 (28.976%)	821 (29.083%)	
IVF + ICSI	128 (7.853%)	128 (7.853%)		126 (8.182%)	127 (8.247%)		393 (13.921%)	379 (13.425%)	

Table results: Mean (SD) Median (Q1–Q3)/N (%) *p*-value: If it is a continuous variable, it is obtained with the Kruskal–Wallis rank sum test; if the count variable has a theoretical number <10, it is obtained using Fisher’s exact probability test. Comparison groups were established by PSM using nearest neighbour matching to adjust the baseline features within the three protocols. BMI, body mass index; IVF, in vitro fertilization; ICSI, intracytoplasmic sperm injection.

**Table 2 jcm-11-06506-t002:** Neonatal outcomes for Natural cycle and Stimulation cycle frozen embryo transfer (FET).

Outcome	Natural Cycle	Stimulation Cycle	Crude OR	P1	Adjusted OR	P2
	1630	1630				
Low BW	63 (3.865%)	72 (4.417%)	1.149 (0.814, 1.624)	0.429	1.152 (0.816, 1.628)	0.422
Preterm	88 (5.399%)	106 (6.503%)	1.219 (0.911, 1.631)	0.183	1.215 (0.908, 1.627)	0.190
Very low BW	7 (0.429%)	9 (0.552%)	1.287 (0.478, 3.465)	0.616	1.284 (0.477, 3.457)	0.621
Very Preterm	8 (0.491%)	12 (0.736%)	1.504 (0.613, 3.688)	0.370	1.489 (0.606, 3.654)	0.385
SGA	66 (4.049%)	94 (5.767%)	1.450 (1.050, 2.002)	0.023 *	1.477 (1.069, 2.041)	0.018 *
LGA	246 (15.092%)	227 (13.926%)	0.910 (0.749, 1.106)	0.345	0.909 (0.747, 1.107)	0.343
IPD	70 (4.294%)	102 (6.258%)	1.488 (1.089, 2.033)	0.012 *	1.511 (1.105, 2.067)	0.010 *
Pre-eclampsia	3 (0.184%)	9 (0.552%)	3.011 (0.814, 11.142)	0.082	3.091 (0.833, 11.473)	0.092
Placental abruption	1 (0.061%)	3 (0.184%)	3.004 (0.312, 28.906)	0.625	2.887 (0.299, 27.862)	0.359

Table results: Mean (SD) Median (Q1–Q3)/N (%), P1 for Crude OR while P2 for Adjusted OR. * Statistically significant, with *p* < 0.05. BW, birthweight; GA, gestational age; S, small; L, large; IPD, ischemic placental disease.

**Table 3 jcm-11-06506-t003:** Neonatal outcomes for Natural cycle and HRT cycle frozen embryo transfer (FET).

Outcome	Natural Cycle	HRT Cycle	Crude OR	P1	Adjusted OR	P2
	1540	1540				
Low BW	59 (3.831%)	68 (4.416%)	1.160 (0.812, 1.656)	0.415	1.159 (0.812, 1.654)	0.418
Preterm	81 (5.260%)	111 (7.208%)	1.399 (1.041, 1880)	0.026 *	1.398 (1.039, 1.880)	0.027 *
Very low BW	7 (0.455%)	4 (0.260%)	0.570 (0.167, 1.952)	0.371	0.577 (0.168, 1.975)	0.381
Very Preterm	8 (0.519%)	8 (0.519%)	1.000 (0.374, 2.671)	1.000	0.997 (0.373, 2.665)	0.996
SGA	60 (3.896%)	62 (4.026%)	1.035 (0.720, 1.486)	0.853	1.032 (0.718, 1.484)	0.865
LGA	242 (15.714%)	234 (15.195%)	0.961 (0.790, 1.168)	0.690	0.959 (0.787, 1.168)	0.679
IPD	64 (4.156%)	74 (4.805%)	1.164 (0.827, 1.639)	0.384	1.164 (0.826, 1.640)	0.386
Pre-eclampsia	3 (0.195%)	13 (0.844%)	4.362 (1.240, 15.337)	0.022 *	4.408 (1.252, 15.512)	0.021 *
Placental abruption	1 (0.065%)	1 (0.065%)	1.000 (0.062, 16.002)	1.000	1.035 (0.064, 16.668)	0.981

Table results: Mean (SD) Median (Q1–Q3)/N (%), P1 for Crude OR while P2 for Adjusted OR. * Statistically significant, with *p* < 0.05. BW, birthweight; GA, gestational age; S, small; L, large; IPD, ischemic placental disease.

**Table 4 jcm-11-06506-t004:** Neonatal outcomes for Stimulation cycle and HRT cycle frozen embryo transfer (FET).

Outcome	Stimulation Cycle	HRT Cycle	Crude OR	P1	Adjusted OR	P2
	2823	2823				
Low BW	124 (4.392%)	148 (5.243%)	0.830 (0.650, 1.060)	0.136	0.830 (0.650, 1.060)	0.136
Preterm	208 (7.368%)	230 (8.147%)	0.897 (0.738, 1090)	0.274	0.898 (0.738, 1.092)	0.280
Very low BW	15 (0.531%)	17 (0.602%)	0.882 (0.439, 1.769)	0.723	0.883 (0.440, 1.772)	0.726
Very Preterm	24 (0.850%)	32 (1.134%)	0.748 (0.439, 1.273)	0.284	0.749 (0.440, 1.275)	0.286
SGA	136 (4.818%)	139 (4.924%)	0.977 (0.767, 1.245)	0.853	0.981 (0.769, 1.250)	0.874
LGA	455 (16.118%)	474 (16.791%)	0.952 (0.827, 1.096)	0.495	0.956 (0.829, 1.101)	0.531
IPD	147 (5.207%)	163 (5.774%)	0.896 (0.713, 1.127)	0.350	0.898 (0.714, 1.130)	0.359
Pre-eclampsia	13 (0.461%)	25 (0.886%)	0.518 (0.264, 1.014)	0.055	0.519 (0.265, 1.016)	0.056
Placental abruption	3 (0.106%)	6 (0.213%)	0.499 (0.125, 1.999)	0.327	0.502 (0.125, 2.012)	0.331

Table results: Mean (SD) Median (Q1–Q3)/N (%), P1 for Crude OR while P2 for Adjusted OR. BW, birthweight; GA, gestational age; S, small; L, large; IPD, ischemic placental disease.

## Data Availability

The data presented in this study are available on request from the corresponding author. The data are not publicly available due to privacy or ethical.

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
