# Peer review of "Effects of Different Endometrial Preparation Regimens during IVF on Incidence of Ischemic Placental Disease for FET Cycles"

_jcm, 2022, doi:10.3390/jcm11216506_

Round 1

Reviewer 1 Report

Dear Authors

I was very pleased to read the article "Effects of different endometrial preparation regimens during IVF on the incidence of ischemic placental disease for FET cycles", however

1. I can't see the tables in the text (please send tchem to me);

2. Remove "References 317"

Besides, the article is accurate with the correct breakdown of the research article.

Author Response

Reviewer1

1.I can't see the tables in the text (please send them to me);

Response: Thanks for your reminding. Tables are already added in the Results part of the text (Page 5-8, Line 193-219).

2.Remove "References 317".

Response: We have already removed it (Page 11, Line 379).

Reviewer 2 Report

The work is based on impressive material, which is a definite advantage. However, I have a few fundamental comments:
Paragraph p. 2, rows 71 - 81 should be singled out as the purpose of the work.
Pg. 2, row 79: Instead of "receiving" it would be better to use "undergoing"
Pg. 3, chapter "Endometrial preparation procedures": A fundamental flaw of the paper is that the endometrial preparation protocol was not decided randomly. Among patients who menstruated irregularly, and were therefore more likely to use ovulation stimulation or HRT for preparation, there was likely a higher rate of PCOS cases accompanied by metabolic syndrome, which in turn predisposes to the development of preeclampsia in pregnancy. This should be included possibly in the discussion as a weakness of the paper.
pp. 3, row 117 instead of "Duupston" probably meant "Duphaston"
pp. 3, row 134: What doses of progesterone suppositories were used? and what does "yellow oral femoral tablet" mean? (trade name of the preparation and manufacturer).
Pages 4 - 6: Baseline Characteristics of included patients and Table 1 should be placed in the Patients and methods section. It does not seem likely that ovulatory dysfunction occurred with the same frequency in the natural and stimulated cycles or HRT groups (note on "Endometrial preparation procedures" chapter as above).
Page 9, rows 318 - 328: Conclusions should be more concise.

Author Response

Dear Reviewers and Editors,

First of all, thank you very much for taking the time out of your busy schedule to read and revise my article. Thank you for your valuable suggestions. You have made a comprehensive correction to the structure, content, research methods and results of my paper. It played a very important role in improving the quality of my papers.

I carefully studied the comments of the reviewers and made careful revisions point to point to the paper according to the recommendations, as follows:

Reviewer2
1.Paragraph p. 2, rows 71 - 81 should be singled out as the purpose of the work.
Response: This section has been listed as a separate section (Page 2, Line 76).

2.Pg. 2, row 79: Instead of "receiving" it would be better to use "undergoing"

Response: We have replaced “receiving” with “undergoing” (Page 2, Line 86).

3.Pg. 3, chapter "Endometrial preparation procedures": A fundamental flaw of the paper is that the endometrial preparation protocol was not decided randomly. Among patients who menstruated irregularly, and were therefore more likely to use ovulation stimulation or HRT for preparation, there was likely a higher rate of PCOS cases accompanied by metabolic syndrome, which in turn predisposes to the development of preeclampsia in pregnancy. This should be included possibly in the discussion as a weakness of the paper.

Response: We have corrected for common confounding factors by the statistical method of PSM and described them in the article (Page 3, Line 103-108), which maintains no difference in baseline between paired groups and thus guarantees the confidence of the results of the study. PCOS was included in the item of ovulation dysfunction for PSM correction (Page 5, Table 1). It is true that not all confounding factors can be fully taken into account, so it has been further explained in the limitation section (Page 10, Line 324-328).

4.pp. 3, row 117 instead of "Duupston" probably meant "Duphaston"

Response: We have replaced “Duupston” with “Duphaston” (Page 3, Line 122).

5.pp. 3, row 134: What doses of progesterone suppositories were used? and what does "yellow oral femoral tablet" mean? (trade name of the preparation and manufacturer).

Response: We have added specific doses of progesterone suppositories, and replaced “femoral” with “Femoston” (Page 3, Line 140-142).

6.Pages 4 - 6: Baseline Characteristics of included patients and Table 1 should be placed in the Patients and methods section. It does not seem likely that ovulatory dysfunction occurred with the same frequency in the natural and stimulated cycles or HRT groups (note on "Endometrial preparation procedures" chapter as above).

Response: In this study, the propensity score matching (PSM) statistics method was applied to correct for the confounding bias in the sample baseline in a 1:1 ratio. We used the endometrium preparation protocol as the dependent variable with the baseline difference term in the unmatched case as the predictor. Data from different subgroups were matched in pairs, thus clarifying the true outcome differences between different endometrial preparation protocols. We have already introduced this statistical technique in the Methods chapter of the article (Page 3, Line 103-108). Ovulation dysfunction as one of the confounding factors in baseline has been corrected by the PSM method (Page 5, Table 1).

7.Page 9, rows 318 - 328: Conclusions should be more concise

Response: The Conclusion chapter has been appropriately deleted (Page 10, Line 350-358).

Reviewer 3 Report

Wang et al conducted a retrospective cohort study and compared pregnancy outcomes especially incidence of ischemic placental disease in frozen embryo cycles. They analyzed three groups depending on the endometrial preparation regimens including natural cycle FET, stimulation cycle and hormone replacement therapy cycle. They found overall natural cycle FET pregnancies had less incidence of pregnancy complications specifically IPD, small for gestational age compared to stimulation cycles and had less incidence of preeclampsia and preterm delivery compared to hormone replacement cycles. There were no differences between stimulation and hormone replacement cycles. The authors conclude that natural cycles could lower the risk of pregnancy complications in selected FET cycles.

I want to congratulate authors for their hard work. This is a clinically very important study. After the review of the manuscript these are my suggestions:

1-      Overall format of the manuscript is well arranged however I suggest the authors correct the spelling, punctuations, format of the citations

2-      The whole manuscript needs a language revisions.

3-      Abstract is well written

4-      Introduction is relevant and relatively short. I suggest authors revise the language and clarify better the third paragraph of the introduction section

5-      Please revise the language and define the methodology especially the endometrial preparation procedures section.

6-      Tables and results section are easy to read and understand

7-      Discussion is overall well written. I suggest authors revise and clearly explain the third paragraph. There is some hypothetical explanation in the second part of the third paragraph as “ Here we focus on …..” and  I suggest this part is important and is explained better to send the correct communication to the author.

Thank you

Author Response

Dear Reviewers and Editors,

First of all, thank you very much for taking the time out of your busy schedule to read and revise my article. Thank you for your valuable suggestions. You have made a comprehensive correction to the structure, content, research methods and results of my paper. It played a very important role in improving the quality of my papers.

I carefully studied the comments of the reviewers and made careful revisions point to point to the paper according to the recommendations, as follows:

Reviewer3

1.Overall format of the manuscript is well arranged however I suggest the authors correct the spelling, punctuations, format of the citations

Response: Article details have been corrected.

2.The whole manuscript needs a language revisions

Response: The article has been polished by professional institutions.

3.Abstract is well written

Response: Thanks for the appraise.

4.Introduction is relevant and relatively short. I suggest authors revise the language and clarify better the third paragraph of the introduction section

Response: The third paragraph of the introduction had been revised (Page 2, Line 54-62).

5.Please revise the language and define the methodology especially the endometrial preparation procedures section.

Response: The language of the endometrial preparation procedures chapter has been revised (Page 3, Line 109-143).

6.Tables and results section are easy to read and understand

Response: Thanks for the appraise.

7.Discussion is overall well written. I suggest authors revise and clearly explain the third paragraph. There is some hypothetical explanation in the second part of the third paragraph as “Here we focus on …..” and  I suggest this part is important and is explained better to send the correct communication to the author.

Response: The third paragraph of the Discussion section has been further interpreted (Page 8, Line 247-258).

Finally, thank you again for your guidance and for reviewing and revising my paper again. I hope that under your guidance, I can complete this excellent paper, and sincerely hope that my paper can be published in your journal.

Round 2

Reviewer 1 Report

The article was supplemented.